# Childhood Trauma and Stressful Life Events Are Independently Associated with Sleep Disturbances in Adolescents

**DOI:** 10.3390/bs9100108

**Published:** 2019-10-10

**Authors:** Suman K.R. Baddam, Rene L. Olvera, Craig A. Canapari, Michael J. Crowley, Douglas E. Williamson

**Affiliations:** 1Yale Child Study Center, Yale School of Medicine, New Haven, CT 06510, USA; Michael.Crowley@yale.edu; 2Department of Psychiatry, University of Texas Health Science Center at San Antonio, San Antonio, TX 78229, USA; olverar@uthscsa.edu; 3Division of Pulmonary Medicine, Department of Pediatrics, Yale School of Medicine, New Haven, CT 06510, USA; craig.canapari@yale.edu; 4Duke Psychiatry & Behavioral Sciences, Duke University School of Medicine, Durham, NC 27710, USA; douglas.williamson@duke.edu

**Keywords:** childhood trauma, stressful life events, sleep patterns, anxiety, depression, adolescence

## Abstract

Adolescence is a critical developmental period associated with an increase in stress, the appearance of anxiety and depressive symptoms, and changes in sleep patterns. Even though the disruption of sleep patterns in stress and anxiety and depressive disorders is well known, the independent effects of childhood trauma and stressful life events on sleep patterns are less understood. We tested the independent effects of stress (childhood trauma and stressful life events) while controlling for anxiety and depression on adolescent sleep patterns. Seven hundred fifty-two adolescents (age 12–15 years) completed self-report questionnaires about childhood trauma, stressful life events, anxiety, and depression. Four sleep factors identifying movement during sleep, sleep regularity, sleep disturbances, and sleep pressure were extracted in the principal component analysis of sleep questions. Both childhood trauma and recent stressful life events were significantly associated with sleep disturbances before and after controlling for anxiety and depression.

## 1. Introduction

Adolescence is a critical transitional period associated with an increase in stress, significant social readjustments [1], and high reactivity to stress [2]. Stress in the form of childhood trauma is common [3] and associated with the development of mental health disorders [4,5,6], lower school engagement [7], poor health outcomes [6], and high suicide rate [5]. Stress in the form of stressful life events is common in adolescence and impacts psychological functioning [8,9], mental health symptoms, and substance use [10]. Severe stressful events in childhood are associated with anxiety and depression in adolescents [11,12,13,14]. A study that assessed both childhood trauma and stressful life events showed that childhood trauma, but not stressful life events, were associated with developing anxiety and depressive symptoms [15,16]. It is possible that severe childhood trauma has a more adverse impact on well-being and differentially impacts the development of mental health symptoms [15,16,17,18].

Childhood trauma such as sexual abuse and physical abuse are associated with multiple sleep disturbances such as difficulty falling asleep, staying asleep, and increased activity during sleep [19,20]. Similarly, stressful life events are associated with insomnia [21,22]. As the number of adverse childhood events increases, so too do sleep disturbances, suggesting a graded relationship between stressful life events and sleep. The specific association of childhood trauma with sleep patterns may be impacted by adolescent mental health disorders as 31.9% of adolescents report an anxiety disorder and 14% a mood disorder [23]. Anxiety and depression consistently present with sleep problems that include obtaining less sleep, trouble falling asleep [24,25], and irregular sleep patterns [26]. The directionality of sleep and mental health symptoms is unclear as sleep disturbances are known to precede [27,28,29] or accompany the appearance of anxiety and depressive symptoms. Dysregulated Hypothalamus—Pituitary—Adrenal (HPA) axis along with altered feedback mechanisms have been identified in stressful events [30], anxiety [31], depressive symptoms [32], and disruption of sleep patterns [33]. Childhood trauma and stress impacts the limbic system and activates the HPA axis. Heightened vigilance and arousal after trauma have been implicated in sleep disturbances [34]. However, the exact pathophysiology underlying the presentation of sleep disturbances in stress, anxiety, and depression are less well understood.

The comorbid association of sleep irregularities in stress and mental health symptoms has not been sufficiently studied. Wang et al. analyzed the association of insomnia with childhood adversity in the national comorbidity survey, a survey of childhood trauma and sleep disturbances in the adolescent United States population. They identified that childhood trauma and insomnia had a dose-response relationship previously discovered in adults [35]. However, the research did not control for mental health symptoms and milder stress of stressful life events, as they are associated with sleep disturbances. In a study controlling for mental health disorders, female adolescents who suffered sexual abuse continued to have sleep disturbances after controlling for PTSD and depression [36]. Another study of adults that assessed both stressful life events and depression showed that stressful life events in the preceding four months were associated with sleep disturbances in depressed adults but not controls [37]. Taken together, even though sleep disturbances are common in childhood trauma and anxiety and depressive symptoms, previous studies primarily focused on childhood trauma and did not adequately control for mental health symptoms. The separate contribution of childhood trauma and stressful life events on mental health [15,16,17,38] are known but that on sleep patterns has not been investigated. Herein, we explored the effects of childhood trauma and stressful events in the past year on sleep patterns after controlling for anxiety and depression. Building upon the current literature of the effects of childhood stress and anxiety and depressive symptoms, we investigated if childhood trauma and stressful life events are independently associated with altered sleep patterns (movement during sleep, regularity of sleep patterns, and disturbances of sleep) in adolescence.

## 2. Materials and Methods

### 2.1. Participants and Procedures

Seven hundred fifty-two adolescents between the ages of 12 years and 14 years, 11 months, were recruited from the greater San Antonio area for a longitudinal study to evaluate the development of alcohol use disorders and depressive disorders. Initial screening excluded those with a diagnosis of Autism Spectrum Disorder and those with an IQ less than 80 based on prior testing or estimations. Subjects with floating metallic objects and dental braces were excluded because of the MRI component of the study. The adolescent participant and their guardian were invited to come for an on-site visit to fill out questionnaires assessing sleep patterns, childhood trauma, stressful life events, and anxiety and depressive symptoms. We are presenting the analyses from the baseline questionnaires. The study was approved by the Institutional Review Board at the University of Texas Health Science Center at San Antonio (Institutional Review Board registered codes IRB00000553, IRB00002691, IRB00002692, and IRB00009608).

### 2.2. Measures

Sleep patterns: Sleep patterns were extracted from eleven sleep questions in the Dimensions of Temperament Survey (DOTS) and four questions in the Youth Self Report (YSR). The Dimensions of Temperament Survey is a widely used instrument developed to measure temperament [39], and the Youth Self Report is a questionnaire commonly used for assessing emotional and behavioral disorders [40]. Sleep questions extracted from the Dimensions of Temperament Survey [39] included “I move a great deal in my sleep” that assessed movement during sleep, “I usually get the same amount of sleep each night” assessed regularity of sleep, and “I take a nap, rest or break at the same time every day” assessed the pressure to sleep, each question rated “Usually false” scored “1,” “More false than true” scored “2,” “More true than false” scored “3,” and “Usually true” scored “4.” The sleep questions from DOTS evaluated the subjective change in sleep patterns related to regularity of bedtime and wake time, movement during sleep and moving in bed, and timing of naps. Sleep questions in the Youth Self Report included “I sleep less than most kids,” assessed disturbances of sleep, and “I sleep more than most kids during day and/or night” were rated “not true,” scored “0,” “Sometimes or Somewhat true” scored “1,” and “very true” scored “2.” The sleep questions from YSR assessed if the adolescents obtained less sleep, had trouble sleeping, needed more sleep and experienced nightmares. Sleep questions from the Youth Self Report and Dimensions of Temperament Survey were used in prior studies for assessing sleep patterns [41,42,43,44].

Childhood trauma: Childhood trauma was assessed by the Childhood Trauma Questionnaire (CTQ). The CTQ is a 28-item validated instrument to measure physical, sexual, and emotional trauma in childhood [45,46,47]. CTQ has high internal consistency (α = 0.95; 0.63–0.95) and test-retest reliability (Intraclass Correlation Coefficient = 0.88) [47]. The questions assessed components of childhood stress such as physical abuse (i.e., I believe that I was physically abused), sexual abuse (i.e., Someone molested me), emotional abuse (i.e., People in my family said hurtful or insulting things to me), emotional neglect (i.e., I didn’t have enough to eat), and physical neglect. Each question was rated “never true” scored “1,” “rarely true” scored “2,” “sometimes true” scored “3,” “often true” scored “4,” and “very often true” scored “5.” The scores for each of the subscales ranged from 5 to 25 and were summed to obtain the total score. The cutoff scores for various subscales of abuse were the following: physical abuse, 10 or higher; sexual abuse, 8 or higher; emotional abuse, 13 or higher; emotional neglect, 15 or higher; physical neglect, 10 or higher [46]. We used the total score for further analyses.

Stressful life events: Stressful life events in the past year were assessed by the Stressful Life Events Schedule (SLES). SLES is an 80-item stress measure validated to measure objective and subjective stress from life events experienced by adolescents in the past 12 months with a test-retest reliability k = 0.68 (95% CI, 0.64–0.72) and inter-rater reliability for objective threat (k = 0.67 ranged from 0.58–0.89). The stress questions assessed life events at school (i.e., I changed schools), job (i.e., I had problems at my job), family (i.e., My family had money problems), legal (i.e., I was a victim of a crime) and personal events (i.e., I had relationship problems with my boyfriend/girlfriend). Each question was rated “not at all” scored “0,” “a little” scored “1,” “somewhat” scored “2” and “lot” scored “3.” Subjective stress score was obtained from the self-report score, whereas the objective stress score was calculated from the objective threat scores given by neutral raters [48].

Anxiety symptoms: Anxiety symptoms were assessed by Screen for Child Anxiety Related Emotional Disorders (SCARED). The SCARED self-report questionnaire is a 41-item instrument for measuring anxiety symptoms in children and adolescents and has high internal consistency (α = 0.93; 0.70 to 0.90) and test-retest reliability (intraclass correlation coefficients = 0.86, 0.70–0.90 for five subfactors and anxiety disorders in the scale). The scale assesses DSM IV-based symptoms of Generalized Anxiety Disorder, Separation Anxiety Disorder, Panic Disorder, Social Phobia, and School Phobia and calculates a total anxiety score for the past three months [7]. Each question was rated on a 3-point scale: “almost never” scored “0,” “sometimes” scored “1” and “often” scored “2.” A total score was calculated from summing individual questions. The scores of the SCARED questionnaire have a range from 0 to 69. A total score of >/=25 in clinical samples is suggestive of an anxiety disorder [49,50].

Depressive symptoms: Depressive symptoms were assessed by the Mood and Feelings Questionnaire Child version (MFQ-C) [51,52]. The Mood and Feelings Questionnaire is a 34-item instrument that measures depressive symptoms in the prior two weeks in the age group 8–18 years and has a high internal consistency (α = 0.90) and test-retest reliability (intraclass correlation coefficient = 0.75). The questions assess depressive symptoms based on DSM III criteria on a three-point scale: “not true” scored “0,” “sometimes true” scored “1,” and “true” scored “2.” The depressive symptoms score was calculated after removing the two sleep questions from the Mood and Feelings Questionnaire. The score of the Mood and Feelings Questionnaire has a range from 0 to 64. The cutoff scores for MFQ used in previous studies is as follows: MFQ < 20 as low, MFQ 20–34 as medium, and MFQ > 34 as high for depression. Wood et al. identified a cutoff score of 27 that represented Major Depression [53].

Parental education, race, and education were assessed by a demographic questionnaire. Parental education was assessed by questions to assess the highest level of education of the parent; less than 9th grade, 9th–12th grade, high school graduate, some college, associate’s degree, bachelor’s degree, and graduate or professional degree. Pubertal status was assessed by tanner staging. Race was assessed by the following categories: White Non-Hispanic, Hispanic, Black, and others.

### 2.3. Statistics

The data was analyzed with IBM Statistical Package for the Social Sciences (SPSS) (Version 26.0). Initially, principal component analysis of the 15 sleep questions was used to extract the sleep factors. Subsequently, General Linear Model Multivariate Analysis of Variance (MANOVA) was utilized to identify the effects of childhood trauma and stressful life events on sleep factors (dependent variables) after controlling for gender, parental education, and psychiatric symptoms (anxiety and depressive symptoms). The effect sizes of the stress measures and psychiatric symptoms on MANOVA’s are represented as Partial Eta Squared (ηp 2) [54]. The estimates of the direction of effect size are represented in t-statistic.

## 3. Results

The demographics of the sample are in Table 1.

Demographic characteristics of the sample:

Note that 256/752 (34%) of the adolescents were of Hispanic origin. The Hispanic or Latino ethnicity comprises 46% of the population in the San Antonio area. Participants were similarly matched for age, race, and parental education between the genders. Females scored significantly higher subjective stress t (750) = −5.05, *p* ≤ 0.001, and objective stress t (750) = −4.89, *p* ≤ 0.001 on SLES and higher anxiety symptoms t (750) = −6.84, *p* ≤ 0.001 and depressive symptoms t (750) = −4.40, *p* ≤ 0.001) than males.

Also, 48/752 (6.2%) of the adolescents met the criteria for emotional abuse, 44/752 (5.9%) met the criteria for physical abuse, 23/752 (3.1%) met the criteria for sexual abuse, 35/752 (4.7%) met the criteria for emotional neglect, and 58/752 (7.7%) met the criteria for physical neglect. In addition, 42/752 (5.6%) had a score greater than 27 on the Mood and Feelings Questionnaire, and 164/752 (21.8%) had a score greater than 25 for anxiety symptoms.

### 3.1. Principal Component Analysis of Sleep Questions

Principal Component Analysis of 15 sleep questions on varimax rotation separated into four factors with an eigenvalue greater than 1 and accounted for 52.09% of the overall variance (see Table 2).

The coefficients with absolute values less than 0.4 were suppressed. Of the four factors, Factor 1 explained 17.41% of the variance and loaded questions that assessed movement while asleep and movement in bed. Factor 2 explained 15.60% of the variance and loaded questions that assessed regularity of wake-up time, bedtime, and regularity of becoming sleepy. Factor 3 explained 10.97% of the variance and loaded questions that assessed sleep disturbances including less sleep, trouble sleeping, and nightmares. Factor 4 explained 8.10% of the variance and loaded questions that assessed the need for more sleep, taking naps, and waking at different times (Table 2). The regression-based factor scores of the four sleep factors were used as dependent sleep scores for further analyses.

### 3.2. Effects of Gender, Age, Race, and Education on Sleep Factors

Initially, we identified the individual effects of the demographic variables of gender, age, race, education, and pubertal status on the identified sleep factors. In these analyses, female gender was significantly associated with higher Factor 1 (F_1, 750_ = 14.05, t = 3.74, *p* ≤ 0.001, ηp 2 = 0.018) and low Factor 2 scores (F_1, 750_ = 7.40, t = −2.72, *p* = 0.007, ηp 2 = 0.010). High parental education was associated with higher Factor 2 scores (regularity of sleep) (F_1, 750_ = 11.12, t = 3.33, *p* = 0.001, ηp 2 = 0.015).

Age, race, and advanced pubertal status were not significantly associated with any of the sleep factors and were not included in further analyses.

Subsequently, we utilized two models to identify the effects of stress. The first model identified the impact of stress (childhood trauma and stressful life events) on the identified sleep factors after controlling for gender and education of parent. The second model identified the association of stress (childhood trauma and stressful life events) with the sleep factors after controlling for anxiety and depressive symptoms, gender, and education of parent.

### 3.3. Correlations between Childhood Trauma, Stressful Life Events, and Anxiety and Depressive Symptom Scores

Childhood trauma score was significantly correlated with the anxiety symptoms score (r (752) = 0.245, *p* ≤ 0.001) and depressive symptoms score (r (752) = 0.457, *p* ≤ 0.001). The stressful life events score had a significant correlation with the anxiety symptoms score (r (752) = 0.323, *p* ≤ 0.001) and depressive symptoms score (r (752) = 0.334, *p* ≤ 0.001). The childhood trauma score and stressful life events score were correlated (r (752) = 0.242, *p* ≤ 0.001) and the anxiety symptoms score was highly correlated with the depressive symptoms score (r (752) = 0.634, *p* ≤ 0.001).

### 3.4. First Model to Identify the Effects of Childhood Trauma and Stressful Life Events on Sleep Patterns After Controlling for Gender and Parent Education

The corrected model was significant for Factor 1 (F_4, 750_ = 6.00, *p* ≤ 0.001, ηp 2 = 0.031), Factor 2 (F_4, 750_ = 7.42, *p* ≤ 0.001, ηp 2 = 0.038) and Factor 3 (F_4, 750_ = 27.65, *p* ≤ 0.001, ηp 2 = 0.129). Factor 4 was not significant (F_4, 750_ = 2.15, *p* = 0.072, ηp 2 = 0.011).

In this model, higher childhood trauma was significantly associated with higher Factor 3 scores (F_1, 750_ = 52.81, t = 7.26, *p* ≤ 0.001, ηp 2 = 0.066) and low Factor 2 scores (F_1, 750_ = 10.09, t = −3.17, *p* = 0.002, ηp 2 = 0.013). Similarly, a higher stressful life events score was significantly associated with higher Factor 3 (F F_1, 750_ = 28.99, t = 5.38, *p* ≤ 0.001, ηp 2 = 0.037), Factor 1 (F_1, 750_ = 5.29, t = 2.30, *p* = 0.022, ηp 2 = 0.007) and Factor 4 scores (F_1, 750_ = 4.16, t = 2.04, *p* = 0.042, ηp 2 = 0.006). The stress measures were not significantly associated with any of the other factors (Table 3).

Female gender in this model was associated with higher Factor 1 scores (F_1, 750_ = 11.15, t = 3.34, *p* = 0.001, ηp 2 = 0.015) and lower Factor 2 scores (F_1, 750_ = 6.57, t = −2.56 *p* = 0.011, ηp 2 = 0.009). High parental education was associated with higher Factor 2 scores (F_1, 750_ = 7.60, t = 2.75, *p* = 0.006, ηp 2 = 0.010). Gender and education were not significantly associated with any of the other factors (Table 3).

### 3.5. Second Model to Identify the Effects of Childhood Trauma and Stressful Life Events on Sleep Factors after Controlling for Anxiety and Depressive Symptoms, Gender, and Education

The corrected model was significant for all the sleep factors: Factor 1 (F_6, 750_ = 7.05, *p* ≤ 0.001, ηp 2 = 0.054), Factor 2 (F_6, 750_ = 5.41, *p* = 0.001, ηp 2 = 0.042), Factor 3 (F_6, 750_ = 39.18, *p* ≤ 0.001, ηp 2 = 0.240) and Factor 4 (F_6, 750_ = 3.42, *p* = 0.002, ηp 2 = 0.027).

Childhood trauma score continued to be significantly associated with higher Factor 3 scores (F_1, 750_ = 10.90, t = 3.30, *p* = 0.001, ηp 2 = 0.014) and lower Factor 2 scores (F_1, 750_ = 9.43, t = −3.07, *p* = 0.002, ηp 2 = 0.013). The Stressful Life Events score was significantly associated with higher Factor 3 scores (F_1, 750_ = 7.76, t = 2.78, *p* = 0.005, ηp 2 = 0.010) and was not associated with the other sleep factors. The effect sizes of stress measures were lower in this model with the inclusion of psychiatric symptoms (Table 4).

High anxiety symptoms in this model were significantly associated with higher Factor 3 scores (F_1, 750_ = 17.30, t = 4.16, *p* ≤ 0.001, ηp 2 = 0.023), higher Factor 1 scores (F_1, 750_ = 10.04, t = 3.17, *p* = 0.002, ηp 2 = 0.013) and high Factor 4 scores (F_1, 750_ = 4.34, t = 2.08, *p* = 0.038, ηp 2 = 0.006). High depressive symptom score was significantly associated with higher Factor 3 scores (F_1, 750_ = 31.09, t = 5.57, *p* = <0.001, ηp 2 = 0.040). Anxiety and depressive symptoms were not significantly associated with the other sleep factors (Table 4).

Female gender in the combined model continued to be significantly associated with lower Factor 2 scores (F_1, 750_ = 8.09, t = −2.84, *p* = 0.005, ηp 2 = 0.011) and higher Factor 1 scores (F_1, 750_ = 5.69, t = 2.38, *p* = 0.017, ηp 2 = 0.008). High parental education continued to be significantly associated with higher Factor 2 scores (F_1, 750_ = 7.62, t = 2.76, *p* = 0.006, ηp 2 = 0.010). Gender and education were not significantly associated with any of the other sleep factors (Table 4).

## 4. Discussion

We evaluated the role of childhood trauma and stressful life events on sleep patterns. We identified four sleep factors—the movement during sleep, the regularity of sleep patterns, sleep disturbances, and sleep pressure. Movement during sleep (Factor 1) originated from questions related to moving in bed and moving during sleep; regularity of sleep (Factor 2) from regularity of bedtime and wake time; sleep disturbances (Factor 3) from questions on sleeping less, trouble sleeping and nightmares; and sleep pressure (Factor 4) from questions on sleeping more, needing naps and waking up at different times. Sleep movement [55,56], regularity of sleep [57,58], sleep disturbances [20], and sleep pressure [59] were used in sleep research to evaluate sleep patterns.

Childhood trauma was significantly associated with sleep disturbances (Factor 3) after controlling for anxiety and depressive symptoms scores. Sleep disturbances (Factor 3) score originated from questions related to nightmares, sleeping less, and difficulty sleeping. These sleep disturbances commonly arise [60] and may persist after the traumatic event. It is interesting to note that the effect size of childhood trauma on sleep disturbances (Factor 3) decreased after anxiety and depressive symptoms were included. The association of childhood adversity with insomnia has been identified elsewhere after controlling for depression [35]. Our results are similar to the findings of Noll et al., where female adolescents who suffered sexual abuse had disturbed sleep after controlling for PTSD and depression [36]. It is pertinent to note that childhood trauma scores in this sample were comparable to a community sample [61] and were lower than an adolescent psychiatric inpatient population [45], suggesting that the association of childhood trauma to sleep exists even at lower levels of trauma.

Similar to childhood trauma, stressful life events in the past year were significantly associated with high sleep disturbance (Factor 3) scores that persist after controlling for anxiety and depressive symptoms. These findings are in line with research that identified that stressful life events are associated with sleep disturbances [21,22]. The effect size of stressful life events on sleep disturbances was lower than childhood trauma, suggesting that the severity of stress plays a critical role in the severity of sleep disturbances. The separate impact of childhood trauma and stressful life events on mental health symptoms [15,16,17,18] was also observed in sleep disturbances in this study. Evaluating stressful life events and severe childhood trauma separately when assessing sleep patterns may be beneficial in delineating the variable impact of stress on sleep patterns. Similarly, controlling for mental health disturbances may be essential as they independently are associated with disturbed sleep. The effect of stress, both childhood trauma and stressful life events, albeit varied, may present through a maladaptive Hypothalamus—Pituitary—Adrenal (HPA) system. Childhood trauma and stressful life events activate the limbic system that, in turn, activate the HPA axis through its projections into the hypothalamus and release the hormones corticotropin-releasing hormone (CRH) and adrenocorticotrophic hormone (ACTH). CRH activates the fast-acting sympathetic-adrenal-medullary system and releases epinephrine in the prefrontal cortex known to increase attention and vigilance, and adrenocorticotrophic hormone (ACTH) releases glucocorticoids from the adrenal cortex. Heightened vigilance and arousal after stress may contribute to sleep disturbances. Laboratory studies have shown that maltreated children and adolescents continue to have increased hypervigilance as they respond to perceived potential social threats [34].

Additionally, anxiety symptoms were significantly associated with high sleep movement (Factor 1), sleep disturbances (Factor 3), high sleep pressure (Factor 4), and depressive symptoms were associated with high sleep disturbances (Factor 3). In previous studies, anxiety and depression symptoms were associated with sleep disturbances in adolescents [20,62]. Evidence suggests that sleep problems and anxiety may manifest together in early adolescence [63] and high movement during sleep was observed in studies of anxiety [64]. The above proposed model of dysregulation of the HPA axis, including the sympathetic system and high glucocorticoids, is implicated in the pathophysiology of insomnia [65], anxiety [31], and depressive symptoms [32]. Future studies should explore the common biological pathways underlying the HPA axis and their variability between stress, sleep, anxiety, and depressive symptoms need to be further explored [20,66].

Female gender is significantly associated with increased movement during sleep, and decreased regularity of sleep. The gender-specific finding of increased disruption of sleep in female adolescents in this study is likely related to the advanced pubertal status of female adolescents when compared to males in this study. Previous studies have shown female-specific increases in sleep problems in adolescents with advanced pubertal status [67]. High parental education was significantly associated with high regularity of sleep. High parental education is associated with earlier bedtime schedules for children [68]. Parentally established bedtimes [69] improve the regularity of sleep by creating regular sleep schedules for adolescents.

This analysis is a secondary analysis of cross-sectional self-reports and has important limitations. We did not have objective measures for sleep and used self-report questionnaires. The four identified sleep factors assessing sleep movement, sleep regularity, sleep disturbance, and sleep pressure were unique. Some degree of association exists between the sleep factors, as sleep habits influence the quality of sleep [70], and poor sleep quality and irregular sleep patterns are associated with increased sleep pressure [71,72]. The sleep labels used were not objectively measured but were reflective of the questions loading into the four factors. Future studies should objectively measure the sleep factors to relate them to stress measures. Subsequent research should identify the differential stress experiences of childhood trauma and stressful life events. The temporal associations between the sleep factors cannot be identified in this analysis because of the cross-sectional nature of the observations. Also, collinearity, although not approaching a high correlation, was observed for stress and anxiety and depressive symptoms, suggesting associations between the answers on reported questionnaires. Future research that includes objective measures of stress may help reduce the associations observed in self-report measures. Lastly, we do not have information about the circadian preferences of the adolescents in our sample, which may be relevant given that adolescents with evening-type sleep preferences have increased depressive and anxiety symptoms [73]. Irregular sleep–wake patterns are associated with later sleep times and naps, highlighting the importance of objectively measuring sleep–wake times in future studies [74].

## 5. Conclusions

In summary, we identified that childhood trauma and stressful life events are independently associated with sleep disturbances, and these effects persist after controlling for anxiety and depressive symptoms in a large cohort of adolescents. This study highlights independent and differential contributions of childhood trauma and stressful life events to sleep patterns previously observed in mental health symptoms. Future research should include objective measures to identify the complex relationships between stress, anxiety, depression, and sleep while focusing on circadian, neurobiological, and social factors [20]. Understanding the common cognitive and physiological pathways and their variability in stress, psychiatric, and sleep symptoms can lead to improved understanding of the development of sleep disturbances in adolescents with stress and affective problems.

## Figures and Tables

**Table 1 behavsci-09-00108-t001:** Demographics of the sample.

	Female	Male	Total/Average ± SD (Standard Deviation)	*T* Test
Number	386	366	752	
Age (years)	13.48 ± 1.01	13.49 ± 1.03	13.48 ± 1.02	t (750) = 0.05, *p* = 0.95
Pubertal status	3.51 ± 0.98	3.27 ± 0.93	3.39 ± 0.96	t (750) = −3.35, *p* = 0.001 **
Race White—nonhispanic	194 (50.3%)	187 (51.1%)	381 (50.7%)	
Hispanic	131 (33.9%)	125 (34.2%)	256 (34.0%)	
Black	12 (3.1%)	13 (3.6%)	25 (3.3%)	
Hispanic and White	33 (8.5%)	22 (6.0%)	55 (7.3%)	
Others	16 (4.2%)	19 (5.1%)	35 (4.7%)	
Education				t (750) = 1.28, *p* = 0.200
Less than 9th grade	1 (0.3%)	2 (0.5%)	3 (0.4%)	
9th to 12th grade (no diploma)	5 (1.3%)	5 (1.4%)	10 (1.3%)	
High School Graduate (including General Education Diploma)	34 (8.8%)	35 (9.6%)	69 (9.2%)	
Some College, no degree	87 (22.5%)	59 (16.1%)	146 (19.4%)	
Associate‘s degree	42 (10.9%)	35 (9.6%)	77 (10.2%)	
Bachelor’s degree	98 (25.4%)	105 (28.7%)	203 (27.0%)	
Graduate or Professional degree	119 (30.8%)	125 (34.2%)	244 (32.4%)	
Childhood trauma score (CTQ)	33.33 ± 8.41	33.64 ± 7.32	33.48 ± 7.90	t (750) = 0.54, *p* = 0.580
Emotional abuse (CTQ)	7.73 ± 3.18	7.24 ± 2.57	7.49 ± 2.91	t (750) = −2.30, *p* = 0.022 *
Physical abuse (CTQ)	6.21 ± 1.83	6.37 ± 1.92	6.29 ± 1.88	t (750) = 1.14, *p* = 0.255
Sexual abuse (CTQ)	5.27 ± 1.45	5.17 ± 0.836	5.22 ± 1.19	t (750) = −1.23, *p* = 0.217
Emotional neglect (CTQ)	8.01. ± 3.47	8.47 ± 3.69	8.23 ± 3.59	t (750) = 1.77, *p* = 0.076
Physical neglect (CTQ)	6.11 ± 1.92	6.40 ± 1.96	6.25 ± 1.95	t (750) = 2.02, *p* = 0.043 *
Stressful life events subjective score (SLES)	140.76 ± 174.35	84.93 ± 122.66	113.59 ± 153.87	t (750) = −5.05, *p* ≤ 0.001 ***
Stressful life events objective score (SLES)	55.52 ± 59.06	36.84 ± 44.03	46.41 ± 53.05	t (750) = −4.89, *p* ≤ 0.001 ***
Anxiety symptoms (SCARED)	19.38 ± 11.25	14.11 ± 9.76	16.81 ± 10.87	t (750) = −6.84, *p* ≤ 0.001 ***
Depressive symptoms (MFQ)	11.86 ± 10.62	8.90 ± 8.27	10.42 ± 9.66	t (750) = −4.40, *p* ≤ 0.001 ***

Significance levels * *p* ≤ 0.05, ** *p* ≤ 0.001, *** *p* ≤ 0.0001.

**Table 2 behavsci-09-00108-t002:** Principal Component Analysis of sleep questions.

	Factor 1 (Questions Assess Movement During Sleep)	Factor 2 (Questions Assess Regularity of Sleep)	Factor 3 (Questions Assess Sleep Disturbances)	Factor 4 (Questions Assess Sleep Pressure)
I move a great deal in my sleep (DOTS)	0.825			
I move a lot in bed (DOTS)	0.862			
In the morning I am in the same place as I fell asleep (DOTS)	−0.597			
I don’t move around much at all in my sleep (DOTS)	−0.835			
I usually get the same amount of sleep each night (DOTS)		0.546		
I get sleepy just about the same time each night (DOTS)		0.601		
When I am away from home, I wake up at the same time each morning (DOTS)		0.686		
No matter when I go to sleep, I wake up at the same time next morning (DOTS)		0.724		
I wake up at the same time on weekends and holidays as on other days of the week (DOTS)		0.673		
I sleep less than most kids (YSR)			0.801	
I have trouble sleeping (YSR)			0.682	
I have nightmares (YSR)			0.409	
I wake up at different times (DOTS)				0.426
I take a nap, rest or break at the same time every day (DOTS)				0.618
I sleep more than most kids during day and/or night (YSR)				0.742

**Table 3 behavsci-09-00108-t003:** Childhood trauma and stressful life events effects on sleep patterns (MANOVAs).

	Factor 1 Movement During Sleep F (*p*)	Factor 2 Sleep Regularity F (*p*)	Factor 3 Sleep Disturbances F (*p*)	Factor 4 Sleep Pressure F (*p*)
Gender	11.15 (*p* = 0.001 ***)	6.57 (*p* = 0.011 *)	0.437 (*p* = 0.509)	0.16 (*p* = 0.689)
Education	1.41 (*p* = 0.235)	7.60 (*p* = 0.006 **)	1.77 (*p* = 0.184)	3.36 (*p* = 0.067)
Childhood Adversity (CTQ Score)	1.53 (*p* = 0.216)	10.09 (*p* = 0.002 **)	52.81 (*p* ≤ 0.001 ***)	0.003 (*p* = 0.957)
Stressful Life Events (SLES Score)	5.29 (*p* = 0.022 *)	0.14 (*p* = 0.700)	28.99 (*p* ≤ 0.001 ***)	4.16 (*p* = 0.042)

MANOVAs, Multivariate Analysis of Variance; F, F-value; *p*, *p*-value; Significance levels * *p* ≤ 0.05, ** *p* ≤ 0.001, *** *p* ≤ 0.0001.

**Table 4 behavsci-09-00108-t004:** Childhood trauma, stressful life events, and mental health symptoms effects on sleep patterns (MANOVAs).

	Factor 1 Movement During Sleep F (*p*)	Factor 2 Sleep Regularity F (*p*)	Factor 3 Sleep Disturbances F (*p*)	Factor 4 Sleep Pressure F (*p*)
Gender	5.69 (*p* = 0.017 *)	8.09 (*p* = 0.005 **)	1.84 (*p* = 0.175)	0.11 (*p* = 0.737)
Education	1.42 (*p* = 0.234)	7.62 (*p* = 0.006 **)	1.85 (*p* = 0.173)	3.46 (*p* = 0.063)
Childhood Adversity (CTQ Score)	0.029 (*p* = 0.864)	9.43 (*p* = 0.002 **)	10.90 (*p* = 0.001 **)	1.13 (*p* = 0.288)
Stressful Life Events (SLES Score)	1.36 (*p* = 0.243)	0.54 (*p* = 0.462)	7.76 (*p* = 0.005 **)	1.13 (*p* = 0.287)
Anxiety symptoms score	10.04 (*p* = 0.002 **)	2.29 (*p* = 0.130)	17.30 (*p* ≤ 0.001 ***)	4.34 (*p* = 0.038 *)
Depressive symptoms score	0.27 (*p* = 0.602)	0.08 (*p* = 0.775)	31.09 (*p* ≤ 0.001 ***)	1.20 (*p* = 0.273)

MANOVAs, Multivariate Analysis of Variance; F, F-value; *p*, *p*-value; Significance levels * *p* ≤ 0.05, ** *p* ≤ 0.001, *** *p* ≤ 0.0001.

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
