# Peer review of "Childhood Trauma and Stressful Life Events Are Independently Associated with Sleep Disturbances in Adolescents"

_behavsci, 2019, doi:10.3390/bs9100108_

Round 1

Reviewer 1 Report

In this paper, the authors compare single timepoint assessments of childhood trauma, stress, and sleep in an adolescent sample aged 12-14. A strength of the study is the moderately large sample and novel analysis of associations of sleep and trauma/stress in this age group. A weakness of the study is the use of purely subjective, non-quantitative measures of sleep, from a series of Likert-type scales, which the authors duly acknowledge in the Discussion. The authors use principal component analysis to generate four sleep-related factors from the series of Likert-type scales. They then report associations between these factors and trauma, stress, and other covariates.  The paper is quite well written and was enjoyable to read.

GENERAL

1)      I wonder why PCA was selected for factoring and if the authors have a justification. PCA is often considered an inferior approach to exploratory factor analysis, because PCA does not discriminate between shared and unique variance (Costello and Osbourne, 2005).

2)      I don’t totally agree with the labels given to the factors. Those in sleep pressure sound more like regularity to me. “I wake up at different times” in particular. The question about naps also seems to me to be not so much about needing naps as about regularity of naps.

3)      It was difficult to discern from the Measures section how sleep regularity was actually assessed. This becomes clear later in the tables, but for a while it was unclear how regularity could be on a true-false scale. It would help to include example questions in text.

4)      p-values should not be represented as 0.000 either in tables or the text. Please use an appropriate inequality. 

5)      Please also be consistent in number of decimal places (or significant figures) for reported values such as partial eta squared.

6)      Why do the stress models not consider adjusting for age or pubertal status? I would imagine stress and trauma exposure may systematically vary with age or pubertal status so could be an important confound. Perhaps the authors already checked this and found it to be non-significant. If so, it would be helpful to report.

7)      It seems the terms “Adverse Childhood Experiences”, “Childhood trauma”, and “Childhood Adversity” are used interchangeably to describe the same scores/predictor. I suggest being consistent.

8)      The authors note that there is no measure of chronotype / sleep timing. I think it is important to note that irregular sleep is robustly associated with later sleep timing in younger adults and adolescents: E.g., see Bei et al. (2016) Sleep Medicine Reviews 28:108-124 and Phillips et al. (2017) Scientific Reports 7:3216.

9)      Given the cross-sectional design I would be careful to avoid potentially causal language, e.g., “the impact of stress on the identified sleep factors”, (line 193).

SPECIFIC

1)      It would be helpful to state the age range of participants in the Abstract.

2)      The sentence on line 37-38 has a grammatical error.

3)      Oxford comma is used inconsistently (sometimes, but usually not).

4)      Sentence on line 46-47 is worded telegraphically and missing a comma.

5)      Line 62: “on” should be “in”.

6)      “Sleep questions extracted from the Dimensions of Temperament Survey [35] evaluated the regularity of bedtime, wake time, moving in sleep, moving in bed, and naps—each rated “Usually false” scored “1”, “More false than true” scored “2”, “More true than false” scored “3” and “Usually true” scored “4”.” Do the authors really mean that they evaluated regularity in each of those five factors (as opposed to the degree of each of those five factors)? For example, I interpret the above passage to possibly mean that the authors evaluated the regularity of moving in sleep (and did not evaluate the degree of moving in sleep), meaning an individual who regularly does not move in sleep and an individual who regularly does move in sleep would be evaluated the same.

7)      I am not sure how to interpret the values presented in Table 3. Since no label is given, I would assume these are estimated effects (lambda values) in the model, but reference to the text suggests they are actually F values. Please make this clear in the caption.

8)      Some of the significant effects in Table 4 are not labeled with the significance levels symbols.

9)      Line 225: Double period.

Author Response

We would like to thank the reviewer for the extremely useful comments in improving the paper.

In this paper, the authors compare single timepoint assessments of childhood trauma, stress, and sleep in an adolescent sample aged 12-14. A strength of the study is the moderately large sample and novel analysis of associations of sleep and trauma/stress in this age group. A weakness of the study is the use of purely subjective, non-quantitative measures of sleep, from a series of Likert-type scales, which the authors duly acknowledge in the Discussion. The authors use principal component analysis to generate four sleep-related factors from the series of Likert-type scales. They then report associations between these factors and trauma, stress, and other covariates.  The paper is quite well written and was enjoyable to read.

GENERAL

In this paper, the authors compare single timepoint assessments of childhood trauma, stress, and sleep in an adolescent sample aged 12-14. A strength of the study is the moderately large sample and novel analysis of associations of sleep and trauma/stress in this age group. A weakness of the study is the use of purely subjective, non-quantitative measures of sleep, from a series of Likert-type scales, which the authors duly acknowledge in the Discussion. The authors use principal component analysis to generate four sleep-related factors from the series of Likert-type scales. They then report associations between these factors and trauma, stress, and other covariates.  The paper is quite well written and was enjoyable to read.

GENERAL

1)         I wonder why PCA was selected for factoring and if the authors have a justification. PCA is often considered an inferior approach to exploratory factor analysis, because PCA does not discriminate between shared and unique variance (Costello and Osbourne, 2005).

Our goal was to use the principal component analysis as a data reduction method to reduce the number of variables to a smaller number of independent composite variables that accounted for the most variance.  We agree with the author that the exploratory factor analysis discriminates between shared and unique variance and provides latent variables. Our goal was not to identify any latent constructs within the variables hence we did not utilize exploratory factor analysis.

2)      I don’t totally agree with the labels given to the factors. Those in sleep pressure sound more like regularity to me. “I wake up at different times” in particular. The question about naps also seems to me to be not so much about needing naps as about regularity of naps.

Thank you for highlighting that the factor labels may not completely reflect the identified factors. In the earlier version of the paper, we did not have factor labels which affected the readability of the paper and the translation of the results in discussion. We acknowledge that the questions in each of the factors may not completely reflect the labels. For that purpose, we changed the Factor 1 (sleep movement) to Factor 1 (questions assessed movement in sleep), Factor 2 (questions assess regularity of sleep), Factor 3 (questions assess sleep disturbances), and Factor 4 (questions assess sleep pressure) in the table. Also, we make an effort to describe the factors whenever possible.

Some of the sleep questions assessed various sleep parameters that included sleep regularity and sleep pressure. We agree that “I wake up at different times” and needing naps may also fit into sleep regularity. However, there is evidence to suggest that taking naps may be reflective of high sleep pressure. Similarly, variable waketimes similarly may be reflective of sleep need and sleep pressure. Because of the collinearity, we continued to utilize sleep pressure for Factor 4 but described the factors wherever possible.

Garbarino, Sergio, Barbara Mascialino, Maria Antonietta Penco, Sandro Squarcia, Fabrizio De Carli, Lino Nobili, Manolo Beelke, Gianni Cuomo, and Franco Ferrillo. 2004. “Professional Shift-Work Drivers Who Adopt Prophylactic Naps Can Reduce the Risk of Car Accidents during Night Work.” Sleep 27 (7): 1295–1302.

Mongrain, V., J. Carrier, and M. Dumont. 2006. “Circadian and Homeostatic Sleep Regulation in Morningness-Eveningness.” Journal of Sleep Research 15 (2): 162–66.

To ensure that the readers understand the limitations of the factor labels, we added the following clarification in line 305“The sleep labels used were not objectively measured but were reflective of the questions loading into the four factors. Future studies should objectively measure the sleep factors to relate them to stress measures.”

3)      It was difficult to discern from the Measures section how sleep regularity was actually assessed. This becomes clear later in the tables, but for a while it was unclear how regularity could be on a true-false scale. It would help to include example questions in text.

Thank you for highlighting the lack of clarity in how the different questions assessed. We included the example questions in the measures section and how they were rated.

The paragraph in the measures section has been changed to

“Sleep questions extracted from the Dimensions of Temperament Survey [35] included “I move a great deal in my sleep" that assessed movement during sleep, "I usually get the same amount of sleep each night" assessed regularity of sleep, and "I take a nap, rest or break at the same time every day" assessed the pressure to sleep with each question rated “Usually false” scored “1”, “More false than true” scored “2”, “More true than false” scored “3” , and “Usually true” scored “4”The sleep questions from DOTS evaluated the regularity of bedtime, wake time, moving in sleep, moving in bed, and naps.—each question rated “Usually false” scored “1”, “More false than true” scored “2”, “More true than false” scored “3” , and “Usually true” scored “4”.  Sleep questions in the Youth Self Report included "I sleep less than most kids" assessed disturbances of sleep and “I sleep more than most kids during day and/or night" were rated “not true” scored “0”, “Sometimes or Somewhat true” scored “1” , and “very true” scored “2”. The sleep questions assessed if the adolescents obtained less sleep, had trouble sleeping, needed more sleep and experienced nightmares.”

4)      p-values should not be represented as 0.000 either in tables or the text. Please use an appropriate inequality. 

Thank you for the highlighting the accurate way of representing p values less than 0.001. We changed the values 0.000 in the document to <0.001

5)      Please also be consistent in number of decimal places (or significant figures) for reported values such as partial eta squared.

We made the decimal places consistent for all the effect sizes and significance levels.

6)      Why do the stress models not consider adjusting for age or pubertal status? I would imagine stress and trauma exposure may systematically vary with age or pubertal status so could be an important confound. Perhaps the authors already checked this and found it to be non-significant. If so, it would be helpful to report.

In our initial analyses, we included age and pubertal status with the sleep symptoms. Age and pubertal status were not significant and were removed from subsequent analyses. We agree with the reviewer and similarly predicted age and pubertal status to have a significant effect. The narrow age range of the sample (12-15 years) and pubertal status may be responsible for the lack of significant findings.

7)      It seems the terms “Adverse Childhood Experiences”, “Childhood trauma”, and “Childhood Adversity” are used interchangeably to describe the same scores/predictor. I suggest being consistent.

Thank you for the comment and the reflection. We agree that the different words used for the same scores and predictor make it confusing. We changed the words to childhood trauma where appropriate.

8)      The authors note that there is no measure of chronotype / sleep timing. I think it is important to note that irregular sleep is robustly associated with later sleep timing in younger adults and adolescents: E.g., see Bei et al. (2016) Sleep Medicine Reviews 28:108-124 and Phillips et al. (2017) Scientific Reports 7:3216.

We agree that measuring chronotype and sleep timing are important in measuring irregular sleep schedules. We added Irregular sleep-wake patterns are associated with later sleep times and naps highlighting the importance of objectively measuring sleep sleep-wake times in future studies [79].  – Line 342

9)      Given the cross-sectional design I would be careful to avoid potentially causal language, e.g., “the impact of stress on the identified sleep factors”, (line 193).

We changed the causal language in the sentence highlighted “The second model identified the effects association of stress (childhood trauma and stressful life events) on with the sleep factors after controlling for anxiety and depressive symptoms, gender, and education.”

SPECIFIC

1)         It would be helpful to state the age range of participants in the Abstract.

We agree with your specific suggestion to make the abstract more readable. We added the ages “(age 12-15 years)” in the abstract section to provide more specific information within the abstract.

2)         The sentence on line 37-38 has a grammatical error.

Thank you for pointing out the error in the sentence. We rephrased the sentence “Stress in the form of stressful life events is are also common in childhood and adolescence and impact psychological functioning [8,9], mental health symptoms and substance use [10].”

3)         Oxford comma is used inconsistently (sometimes, but usually not).

We added the oxford comma wherever applicable to make it more consistent. The added commas are bolded and in teal color.

4)         Sentence on line 46-47 is worded telegraphically and missing a comma.

Thank you for highlighting the telegraphic sentence. We made the following modification.

“Similarly, stressful life events are associated with sleep disturbances such as insomnia [19,20], and higher the number of adverse childhood events experiences, higher the sleep disturbances suggesting a graded relationship between stressful life events and sleep [21,22].”

5)         Line 62: “on” should be “in”.

Changed the on to in as appropriately suggested by the reviewer. The sentence is Even though the disruption of sleep patterns in childhood trauma and anxiety and depressive symptoms is known, the independent contribution of childhood trauma and stressful life events in on childhood on sleep patterns has not been investigated.”

6)         “Sleep questions extracted from the Dimensions of Temperament Survey [35] evaluated the regularity of bedtime, wake time, moving in sleep, moving in bed, and naps—each rated “Usually false” scored “1”, “More false than true” scored “2”, “More true than false” scored “3” and “Usually true” scored “4”.” Do the authors really mean that they evaluated regularity in each of those five factors (as opposed to the degree of each of those five factors)? For example, I interpret the above passage to possibly mean that the authors evaluated the regularity of moving in sleep (and did not evaluate the degree of moving in sleep), meaning an individual who regularly does not move in sleep and an individual who regularly does move in sleep would be evaluated the same.

Thank you for the comments highlighting the lack of clarity of the statement. We expanded the paragraph to include the exact questions from the Dimensions of Temperament Survey and changed the paragraph as below.

“Sleep questions extracted from the Dimensions of Temperament Survey [39] included “I move a great deal in my sleep" that assessed movement during sleep, "I usually get the same amount of sleep each night" assessed regularity of sleep, and "I take a nap, rest or break at the same time every day" assessed the pressure to sleep each question rated “Usually false” scored “1”, “More false than true” scored “2”, “More true than false” scored “3” , and “Usually true” scored “4”The sleep questions from DOTS evaluated the subjective change in sleep patterns related to regularity of bedtime and wake time, moving in movement during sleep and moving in bed, and timing of naps.—each question rated “Usually false” scored “1”, “More false than true” scored “2”, “More true than false” scored “3” , and “Usually true” scored “4”.

7)      I am not sure how to interpret the values presented in Table 3. Since no label is given, I would assume these are estimated effects (lambda values) in the model, but reference to the text suggests they are actually F values. Please make this clear in the caption.

Added captions to Tables 3 and 4

8)      Some of the significant effects in Table 4 are not labeled with the significance levels symbols.

Added the stars to the two significance levels to flag significance.

9)      Line 225: Double period.

Removed the double period and rechecked the whole paragraph.

Reviewer 2 Report

Very interesting article, well done

Author Response

We thank the reviewer for the kind comments and reviewing the article.

Reviewer 3 Report

This is an interesting paper on a  innovative topic.

There is a  limit related  to the  lack of sleep  objective measures  with appropriate sleep tools ( actigraphy)

Author Response

We thank the reviewer for the comments. Certainly, we acknowledge in this current paper, the lack of objective measures of sleep and stress limits the validity of the findings. This paper can be helpful for future research using objective methods of sleep and stress.

Reviewer 4 Report

Behavioral Sciences: Childhood trauma and stressful life events are independently associated with sleep disturbances in adolescents.

Introduction:

·      The introduction sets up the expectation that depression and anxiety are key outcome variables. Yet, in the results they are used as statistical control variables. The introduction would be streamlined if the authors focused on the main outcome (sleep).

·      The manuscript would benefit from the presentation of some clear hypotheses. Linking clear hypotheses to statistical tests will greatly improve this manuscript.

·      In the introduction, the authors state that the independent contribution of childhood trauma and stressful life events on childhood sleep patterns has not been investigated. However, there are published manuscripts that examine these associations in children and adolescents. The authors should highlight how their contribution is different from previously published literature. See Wang et al., 2016 Sleep Medicine as an example.

·      The authors should use the term adolescence instead of childhood throughout the introduction.

·      In general the introduction would benefit from including an overarching framework or theory.

Method

·      The authors should include information about how parental education, race, education, and pubertal status were measured in the method section.

·      The authors should report the psychometrics of the sleep outcome.

·      Did the authors use a total score or clinical cutoffs for the CTQ in their analyses?

Results

·      It would be preferable if the correlation data were presented first.

·      The manuscript would be easier to read if the factor labels were used instead of factor 1, factor 3, etc. in describing the results.

·      Instead of using high and low to describe the results, more specific descriptors would be helpful. For example, higher childhood trauma was associated with greater sleep disturbances.

Discussion

·      The authors could discuss the finding of the model that includes both childhood trauma and current stress in terms of predictive utility.

·      One way to address the high correlation among stress, anxiety and depression would be to center the predictors. The authors could also present partial correlation coefficients.

Author Response

We would like to thank the reviewer for thoughtful comments to enhance the quality and the science in the paper.

Introduction:

The introduction sets up the expectation that depression and anxiety are key outcome variables. Yet, in the results they are used as statistical control variables. The introduction would be streamlined if the authors focused on the main outcome (sleep).

Thank you for the suggestion to streamline the introduction to remove the primary focus on anxiety and depression as they are the control variables. We rewrote the introduction to reduce the emphasis on anxiety and depression as they are variables we are controlling for in the analyses.

The manuscript would benefit from the presentation of some clear hypotheses. Linking clear hypotheses to statistical tests will greatly improve this manuscript.

We rewrote the introduction to ensure that there clear hypotheses building on previous research. The last paragraph of the introduction is below.

“The comorbid association of sleep in stress and mental health symptoms has not been sufficiently studied. Wang et al analyzed the association of insomnia with childhood adversities in the national comorbidity survey, a survey of mental health, stress, and sleep disturbances in the adolescent United States population. They identified childhood trauma and insomnia had a dose-response relationship previously discovered in adults [35]. However, the research did not control for mental health symptoms and milder stress of stressful life events as they are associated with sleep disturbances. One such study In In a study controlling for mental health disorders, female adolescents who suffered sexual abuse continued to have sleep disturbances after controlling for PTSD and depression [36]. Another study of adults that assessed both stressful life events and depression showed controlling for depression in adults identified that at least one disrupting stressful life events in the preceding past four months were associated with sleep disturbances in depressed adults but not controls [37]. Taken together, even though sleep disturbances the disruption of sleep patterns are common in childhood trauma and anxiety and depressive symptoms is known, previous studies primarily focused on childhood trauma and did not adequately control for mental health symptoms. The separate contribution of childhood trauma and stressful life events on mental health [15–17,38] are known but that on sleep patterns has not been investigated. In this study, we investigate the independent association contribution of childhood trauma and stressful life events in on childhood on sleep patterns. Herein, we explored the effects of childhood trauma and stressful events in the past year on sleep patterns after controlling for anxiety and depression. Building upon the current literature of the effects of childhood stress on sleep and anxiety and depressive symptoms, we investigated if childhood trauma and stressful life events are independently associated with altered sleep patterns (movement during sleep, regularity of sleep patterns, and disturbances of sleep) in adolescence”.

In the introduction, the authors state that the independent contribution of childhood trauma and stressful life events on childhood sleep patterns has not been investigated. However, there are published manuscripts that examine these associations in children and adolescents. The authors should highlight how their contribution is different from previously published literature. See Wang et al., 2016 Sleep Medicine as an example.

Thank you for the useful suggestion. We further built the introduction so that it highlights the work by Wang et al in the sleep medicine paper and highlight how this current study adds to the existing literature. The following is the reformatted introduction.

“Wang et al analyzed the association of insomnia with childhood adversities in the national comorbidity survey, a survey of mental health, stress, and sleep disturbances in the adolescent United States population. They identified childhood trauma and insomnia had a dose-response relationship previously discovered in adults [35]. However, the research did not control for mental health symptoms and milder stress of stressful life events as they are associated with sleep disturbances. One such study In In a study controlling for mental health disorders, female adolescents who suffered sexual abuse continued to have sleep disturbances after controlling for PTSD and depression [36].”

The authors should use the term adolescence instead of childhood throughout the introduction.

We used the term adolescence consistently in the introduction where applicable.

In general the introduction would benefit from including an overarching framework or theory.

Thank you for the useful suggestion. We rewrote the introduction to provide a theoretical basis of the effects of severity of stress, mental health symptoms associated with disrupted sleep patterns and the severity of stress playing a role in sleep problems.

Method

The authors should include information about how parental education, race, education, and pubertal status were measured in the method section.

Parental education, race, education were assessed by a demographic questionnaire. Parental education was assessed by questions to assess the highest level of education of the parent; less than 9th grade, 9-12 grade, high school graduate, some college, associate’s degree, bachelor’s degree, and graduate or professional degree. Pubertal status was assessed by tanner staging. Race was assessed by the following categories White Non-Hispanic, Hispanic, black, and others.

The authors should report the psychometrics of the sleep outcome.

We extracted the individual questions from established questionnaires and presented the psychometrics of the questionnaires. Since the extracted questions were used to identify principal components and not develop a specific sleep outcome, we did not calculate the psychometrics.

Did the authors use a total score or clinical cutoffs for the CTQ in their analyses?

Thank you for the clarifying question. We used the total score of the Childhood Trauma Questionnaire in the analyses. We highlighted that we used the total score of the Childhood Trauma Questionnaire.

Results

It would be preferable if the correlation data were presented first.

We moved the correlation data and table to below the demographic results.

The manuscript would be easier to read if the factor labels were used instead of factor 1, factor 3, etc. in describing the results.

We limited using Factor 1, 2, 3 and used the descriptive labels in the discussion. We did not eliminate using the factor labels completely because the factor labels may not be completely explained by the questions extracted.

Instead of using high and low to describe the results, more specific descriptors would be helpful. For example, higher childhood trauma was associated with greater sleep disturbances.

We changed the comparative adjectives to higher and lower. The changed adjectives are highlighted in blue.

Discussion

The authors could discuss the finding of the model that includes both childhood trauma and current stress in terms of predictive utility.

Thank you for the suggestion about using a model. We expanded the introduction and discussion to highlight the importance of including both childhood trauma and stressful life events.

“The separate impact of childhood trauma and stressful life events on mental health symptoms [15–18] was also observed in sleep disturbances in this study. Evaluating stressful life events and severe childhood trauma separately when assessing sleep patterns may be beneficial in delineating the variable impact of stress on sleep patterns. Similarly, controlling for mental health disturbances may be essential as they independently are associated with disturbed sleep. The effect of stress, both childhood trauma and stressful life events, albeit varied may present through a maladaptive Hypothalamus – Pituitary – Adrenal (HPA) system. Childhood trauma and stressful life events activate the limbic system that, in turn, activate the HPA axis through its projections into the hypothalamus and release the hormones corticotropin-releasing hormone (CRH) and adrenocorticotrophic hormone (ACTH). CRH activates the fast acting sympathetic-adrenal-medullary system and releases epinephrine in the prefrontal cortex known to increase attention and vigilance, and adrenocorticotrophic hormone (ACTH) releases glucocorticoids from the adrenal cortex. Heightened vigilance and arousal after stress may contribute to sleep disturbance. Laboratory studies have shown that maltreated children and adolescents continue to have increased hypervigilance as they respond to perceived potential social threats [34].”

One way to address the high correlation among stress, anxiety and depression would be to center the predictors. The authors could also present partial correlation coefficients.

The correlation among the stress, anxiety and depression measures was significant. The correlation coefficients observed were modest (less than 0.457 between childhood trauma and stressful life events, and anxiety and depression) and not high hence we did not center the predictors.
